# Effect of Temperature and Photoperiod Preconditioning on Flowering and Yield Performance of Three Everbearing Strawberry Cultivars

**Rodmar Rivero [1,*], Siv Fagertun Remberg [1], Ola M. Heide [2] and Anita Sønsteby [3]**

[1] Faculty of Biosciences, Norwegian University of Life Sciences, NO-1432 Ås, Norway; siv.remberg@nmbu.no
[2] Faculty of Environmental Sciences and Natural Resource Management, Norwegian University of Life Sciences, NO-1432 Ås, Norway; ola.heide@nmbu.no
[3] NIBIO—Norwegian Institute of Bioeconomy Research, NO-1431 Ås, Norway; anita.sonsteby@nibio.no
* Correspondence: rodmar.rivero.casique@nmbu.no; Tel.: +47-90408434

**Abstract:** Environmental control of flowering in everbearing strawberry is well known, while the optimal commercial raising conditions for high and continuous yield remains unsettled. We exposed freshly rooted plants of cultivars Altess, Favori and Murano to 9 °C, 15 °C, 21 °C and 27 °C, respectively, at two photoperiods for 4 weeks, and assessed flowering and yield performance. Long days at 15–21 °C enhanced flowering, while short days (SD), particularly at 27 °C, decreased flowering. Runner formation was enhanced by SD, being inversely related to flowering. Yields the next season were highest in plants exposed to 15–21 °C, whereas the seasonal harvest distribution varied. In concurrence with earlier reports, the size of the first fruit flush determined the magnitude of the second flush and the length of the off period when little fruit was produced. The large first fruiting flushes of plants exposed to 21 and 27 °C gave particularly long off periods and small second flushes. Moderate first flushes of plants from intermediate temperatures also resulted in a more evenly distributed harvest and the largest yields. Developing flowers and fruits are strong sinks for photosynthates; therefore, the size of the first fruit flush must be compromised to optimize fruit yield and seasonal crop distribution.

**Keywords:** everbearing; *Fragaria × ananassa*; photoperiod; preconditioning; temperature; yield

## 1. Introduction

In contrast to the traditional seasonal flowering (SF) strawberry cultivars which are quantitative short day (SD) plants at intermediate temperatures (18–21 °C) and unable to flower at high temperature (27 °C), the everbearing (EB) cultivars are quantitative long day (LD) plants at intermediate temperatures (18–24 °C) and obligatory LD plants at high temperature (27 °C). However, at low temperatures (<15 °C), both types are day neutral and flower independently of photoperiod [1,2]. These diverse flowering responses have important implications for how the two plant types should be raised and cultivated.

In Europe, the EB strawberries are mainly used for annual production in plastic tunnels. Usually, runner tips are rooted in late July and raised during autumn under outdoor conditions in specialized plant nurseries in The Netherlands, where the production of such ready-to-flower plants has developed into a considerable industry [3]. Under the relatively low temperature conditions in autumn, the plants will initiate flower buds even in SD. After overwintering in cold storage at −1.5 °C, the plants were established in March–April on a tabletop production system for cropping in plastic tunnels or greenhouses. For convenience, and because of the marginal temperature conditions prevailing in the north, strawberry growers in Nordic countries usually buy their plants from The Netherlands. However, Sønsteby et al. [4] recently reported that plants with the same yield potential as

the imported plants could be produced under Nordic temperature conditions according to a slightly modified production protocol.

A serious shortcoming of the European production system is, however, that it does not produce a continuous and stable supply of ripe berries during the harvest season, but rather a series of flowering and fruiting flushes separated by gaps with little or no production [4–7]. The first, and usually largest, fruit flush, which originates from inflorescences produced during plant raising in the previous year [7], is usually followed by an off period of 2–3 or more weeks with little fruit. This causes discontinuous fruit supply and reduced total yields and represents a big challenge in commercial production.

Melis [6] observed that a large first fruit flush produced by large plants with high yield potential was always associated with a long off period, whereas a smaller first flush was combined with a shorter off period. Therefore, he argued that a heavy crop load tended to suppress and delay the initiation of recurrent flower flushes. This was directly supported by the results of Sønsteby et al. [7] who found that both floral initiation and subsequent fruit growth were source-limited in heavily fruiting plants. However, there seemed to be differences in the severity of the problem among cultivars. Thus, the high yielding 'Favori' commonly produces a large first flush, followed by a long off period, whereas the lower yielding 'Murano' has a smaller first fruit flush and a more even distribution of the crop during the fruiting season [7].

This means that the total fruit yield of the EB cultivars is determined by the yield potential established during plant raising, as well as the additional recurrent floral initiation taking place during the cropping season. Accordingly, optimization of fruit yield requires harmonization of the two floral initiation steps (cf. Sønsteby et al. [4]).

Despite the knowledge available on the physiology of flowering of EB strawberries [1], there is need for further investigation on the flower-inducing efficiency of the various temperature and photoperiodic conditions in a range of cultivars under conditions that are relevant for commercial production. This prompted us to perform a controlled environment experiment in which three commercial cultivars were preconditioned at temperatures ranging from 9–27 °C in 10-h and 20-h photoperiods for four weeks during plant raising. Their instant flowering potential, as well as their final yield potential after overwintering in a cold store and cropping in a plastic high tunnel, were assessed.

The aim of the experiment was two-fold: (1) to provide a firm knowledge basis for the flower-inducing efficiency of a range of relevant temperatures and photoperiods in commercial cultivars, and (2) to study the seasonal crop distribution of the treated plants and its relation to total yield.

## 2. Materials and Methods

### 2.1. Plant Material and Cultivation

The plant material used for this experiment was propagated in a greenhouse at the NIBIO Experimental Centre Apelsvoll in southeast Norway (60°40′ N, 10°40′ E). Three commercially available and runner-propagated EB strawberry cultivars (*Fragaria × ananassa* Duch.) were used for the experiment: Altess (Flevo Berry Holding B.V., Ens, The Netherlands), Favori (Flevo Berry Holding B.V., The Netherlands) and Murano (Conzorcio Italiano Vivaisti, C.I.V., Comacchio, Italy). Young runner plants of all cultivars were collected in mid-July from mother plants grown in a plastic tunnel at NIBIO Apelsvoll. All runner plants were rooted directly in 9 cm pots in a peat-based potting compost (Gartnerjord, LOG, Oslo, Norway) mixed with 20% (*v/v*) granulated perlite in a water-saturated atmosphere under a plastic enclosure at 20 h photoperiod and a minimum temperature of 20 °C. On 14 August, all plants were transferred to the phytotron at the Norwegian University of Life Sciences at Ås (59°40′ N, 10°40′ E) and exposed to 10-h short day (SD) and 20-h long day (LD) at temperatures of 9 °C, 15 °C, 21 °C or 27 °C for 4 weeks. In the phytotron, all plants were grown during daytime (8:00 a.m. to 8:00 p.m.) in natural daylight compartments and then moved to adjacent growth rooms from 18:00 h–08:00 h. There, they received either darkness for 14 h (SD) or 10 h low-intensity-light (~7 μmol quanta m$^{-2}$ s$^{-1}$ photosynthetic

photon flux (PPF)) from 70 W incandescent lamps for daylight extension (LD), so that the 4 h dark period was centered around midnight (10:00 p.m. to 2:00 a.m.). The daylight extension amounted to less than 2% of the total daily light radiation, all plants thus receiving nearly the same daily light integral in both photoperiods. In the daylight compartments, an additional 125 µmol quanta m$^{-2}$ s$^{-1}$ were automatically added by high-pressure metal halide lamps (400 W Philips HPI-T) whenever the PPF in the compartments fell below 150 µmol quanta m$^{-2}$ s$^{-1}$ (as on cloudy days). The plant trolleys were positioned randomly in the daylight rooms due to the daily movement in and out of the adjacent photoperiodic treatment rooms. Temperatures were controlled to $\pm1$ °C and a water vapor pressure deficit of 530 Pa was maintained at all temperatures. Throughout the experimental period, the plants were irrigated daily to drip-off with a complete fertilizer solution [electric conductivity 1.2–1.4 mS cm$^{-1}$, 1:1 YaraTera Kristalon™: YaraLiva Kalksalpeter™ (Yara, Oslo, Norway)].

After this four-week preconditioning treatment, all plants were potted in 12 cm pots with a peat based potting compost, and three plants from each replicate and treatment were forced directly in a greenhouse for 10 weeks with 20 h LD at 20 °C for assessment of their instant flowering status. In the greenhouse, the plants received daylight plus a daily mix of artificial light of approx. 150 µmol quanta m$^{-2}$ s$^{-1}$ PPF from 400 W Philips HPI-T metal halide lamps (8:00 a.m. to 6:00 p.m.) plus about 15 µmol quanta m$^{-2}$ s$^{-1}$ light from 70 W incandescent lamps (6:00 p.m. to 10:00 p.m. and 2:00 a.m. to 8:00 a.m.) throughout the forcing period. The rest of the plants were moved outdoors for continued floral initiation and stabilization from 12 September to 24 October 2018. The daily mean temperatures during this period are shown in Figure 1A. Thereafter, one plant from each replicate was forced for 8 weeks under the same conditions as explained above for assessment of flowering and yield potential. Due to the limited number of 'Murano' plants, this treatment had to be omitted for this cultivar. The remaining plants were cold stored in darkness at −1.5 °C during the period 24 October 2018–5 May 2019.

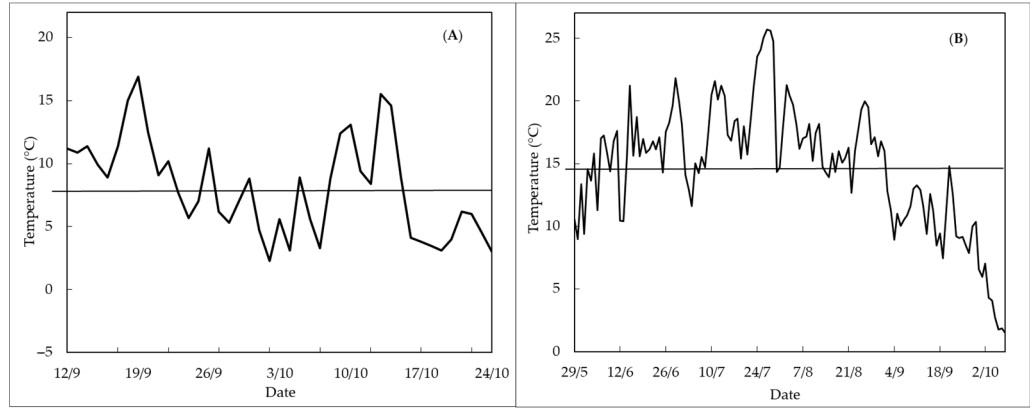

**Figure 1.** Daily mean temperatures during the outdoor period of plant raising in 2018 (**A**), and in the plastic tunnel during the cropping season in 2019 (**B**). The horizontal lines represent the average daily mean for the respective periods.

The cold-stored plants were then cropped in a tabletop system in a plastic high tunnel for assessment of yield and crop distribution. On 5 May 2019, all plants were transplanted into 2.5 L plastic pots (one plant per pot) in a mixture of 80% limed and fertilized peat and 20% granulated perlite. After a 9-day establishment period in an unheated plastic greenhouse under a double layer of fiber cloth, the plants were placed in an open plastic tunnel on 16 May, and drip irrigated with a nutrient solution containing a 1.1 mixture of YaraTera CALCINIT® and YaraTera Kristalon® (Yara, Norway) with electric conductivity of 1.6 mS cm$^{-1}$. The daily mean temperatures in the tunnel during the cropping season are shown in Figure 1B.

*2.2. Experimental Design and Data Observations*

During preconditioning in the phytotron, the experiment was conducted as a randomized block design with three replicates with 10 plants of each cultivar in each treatment. Growth was monitored by weekly registration of the number of leaves, crowns, and runners. At termination of the preconditioning treatments and before forcing or cold storage, the total number of leaves, runners, flower trusses and flowers, and petiole length of the last fully developed leaf were also recorded. During forcing, flowering and growth performance were assessed by weekly recordings of the total number of leaves, runners, flower trusses and open flowers. In the plastic tunnel, the experiment was conducted as a randomized block design with three replicates of 4 plants of each cultivar in each treatment. Ripe berries were harvested 2–3 times per week from 5 July to 2 October. The number and weight of all berries, including unsalable berries, were recorded, as well as the proportion of healthy berries with diameter > 25 mm. In addition, runners were registered and removed throughout the season. At termination of the experiment on 2 October, plant height (measured from base to top of the leaf canopy), number of crowns, leaves per plant and plant fresh weight (excluding runners and roots) were recorded on all plants, as were the number of flowers and berries not reaching maturity.

Experimental data were subjected to analysis of variance (ANOVA) using the MiniTab® v18.1 Statistical Software program package (Minitab Inc., State College, PA, USA). Prior to the analyses, homoscedasticity and normality assumptions were tested (Ryan–Joiner test for normality and Levene's test for homoscedasticity). Percentage values were always subjected to square root transformation before performing the ANOVA.

## 3. Results

*3.1. Flowering and Runnering in the Phytotron*

During the preconditioning treatment in the phytotron, open flowers started to appear after three weeks of treatment (after one week at high temperature in 'Murano'), indicating that the plants were induced to flower before the treatments were started (Figure 2). This is typical for EB strawberries, in which flowers usually appear as soon as the runners are formed [2]. The number of flowers was highest in 'Murano' and increased over time with increases in temperature and photoperiod in all cultivars. Runners emerged after two weeks in all cultivars, and the highest number was observed in 'Favori' at intermediate to high temperatures in both photoperiods. The main effects of temperature and cultivar, as well as the two-factor interaction between cultivar and temperature, were all significant (Appendix A, Table 1).

*3.2. Flowering Potential of the Preconditioned Plants*

Plants from all cultivars and preconditioning treatments started to flower after 2 weeks of forcing, their numbers increasing significantly over time with increases in temperature and photoperiod (Figure 3, Table 2). In all cultivars and at all temperatures, the number of flowers increased in LD, being highest at 15 °C in 'Altess' and 'Murano', and at 21 °C in 'Favori'. Under SD conditions, flowering was strongly delayed at 27 °C in 'Favori' and 'Murano', and to a lesser extent, this was the case at all temperatures in all cultivars. However, after a time span of approximately seven weeks, a new flush of flowers emerged in these plants. Most likely, these flowers were initiated during the forcing treatment in LD at 20 °C. At both forcing times, the main effects of both temperature and photoperiod were significant, with no significant cultivar interactions (Appendix A, Table 2).

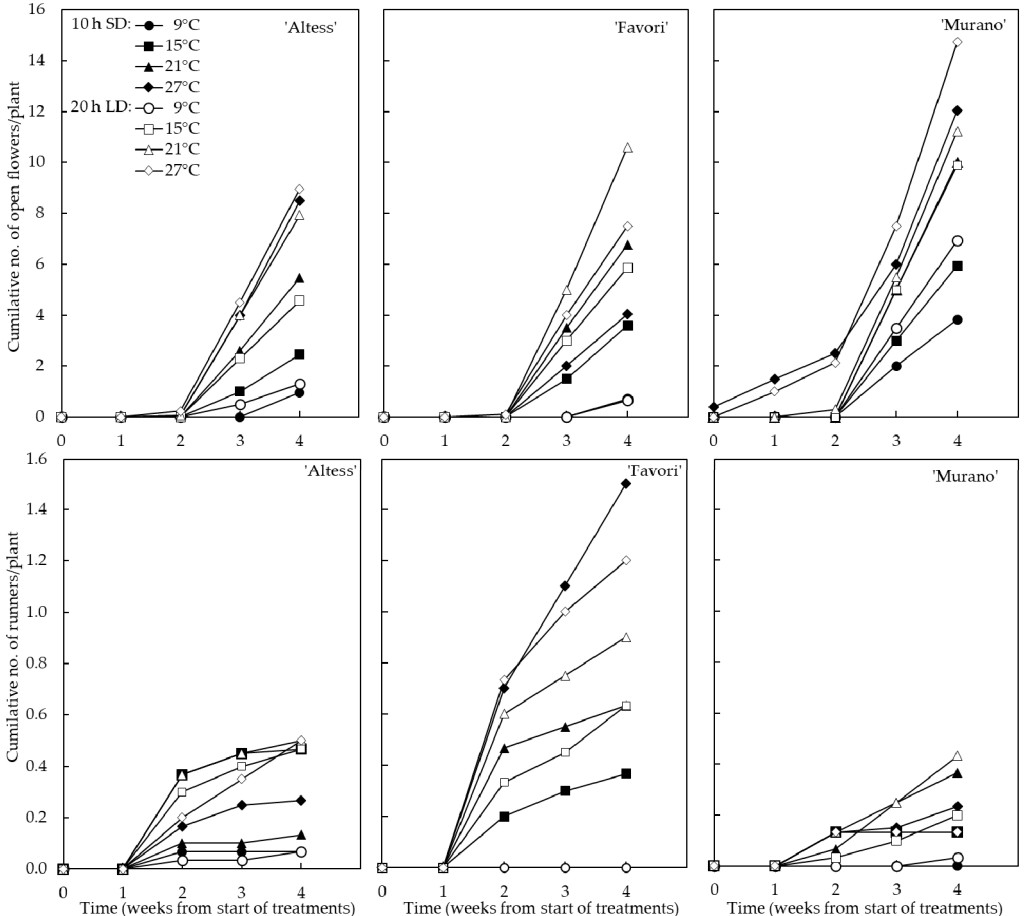

**Figure 2.** Time courses of cumulative flower and runner appearances during the 4-week preconditioning period in three EB strawberry cultivars. Data are the means of three replicates, each with 10 plants of each cultivar.

Runners started to emerge after two weeks of forcing in all cultivars but remained generally low in plants preconditioned in LD. The number of runners was largest in 'Altess' and 'Murano', in which it increased rapidly during the first 6 weeks of forcing in plants preconditioned in SD at intermediate to high temperatures. Due to varying interactions between temperature and photoperiod, and between the environments and cultivars, the main effects of temperature and photoperiod were not always significant but were generally strongest in the first forcing (Table 2). The results revealed an opposite relationship between flowering and runner formation, with flowering being promoted by LD and intermediate temperatures while runner formation was promoted by SD at the same temperature range.

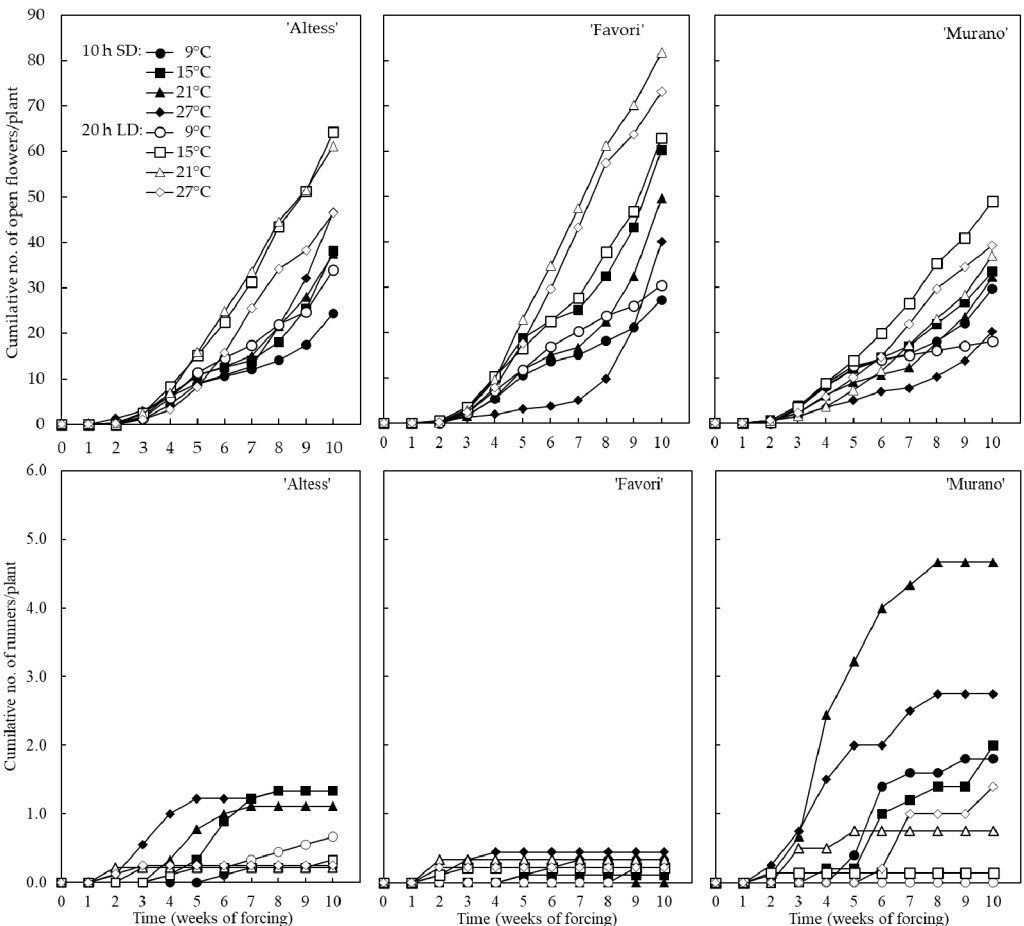

**Figure 3.** Time courses of cumulative flower (top panel) and runner formation in three EB strawberry cultivars as affected by 4 weeks of temperature and photoperiod preconditioning, as indicated during 10 weeks of subsequent forcing in a greenhouse in 20-h LD at 20 °C. The data are means of three replicates, each with three plants of each cultivar.

### 3.3. Plant Flowering Potential in Late Autumn

Time courses of flower and runner emergences during an 8-week forcing period started on 24 October (at the end of the raising period) are shown in Figure 4 and Table 2. Due to an insufficient number of 'Murano' plants, only data for 'Altess' and 'Favori' are available. In all plants of both cultivars, flowers started to emerge after 3 weeks of forcing, while the number of flowers varied widely in the various treatments, being highest after preconditioning in LD and intermediate to high temperatures. In SD, on the other hand, flowering was generally sparse and, under all conditions, flowering started to level off after 3–4 weeks. However, after 7 weeks of forcing, a new flush of flowers emerged in all plants regardless of pretreatment conditions. Since the number of additional flowers increased in parallel and started at the same time in all treatments, they were apparently initiated during the flower-inducing (LD and 20 °C) forcing treatment (cf. Figure 3). Generally, the two cultivars responded in the same way; only the main effects of temperature and photoperiod were statistically significant (Table 2).

Runners started to emerge shortly after the forcing was started in both cultivars but remained low for the first 4–5 weeks (Figure 4). From week 5 onwards, the number of runners increased somewhat again in the sparsely flowering plants preconditioned in SD. The main effect of temperature and the two-factor interaction between temperature and photoperiod were statistically significant (Table 2).

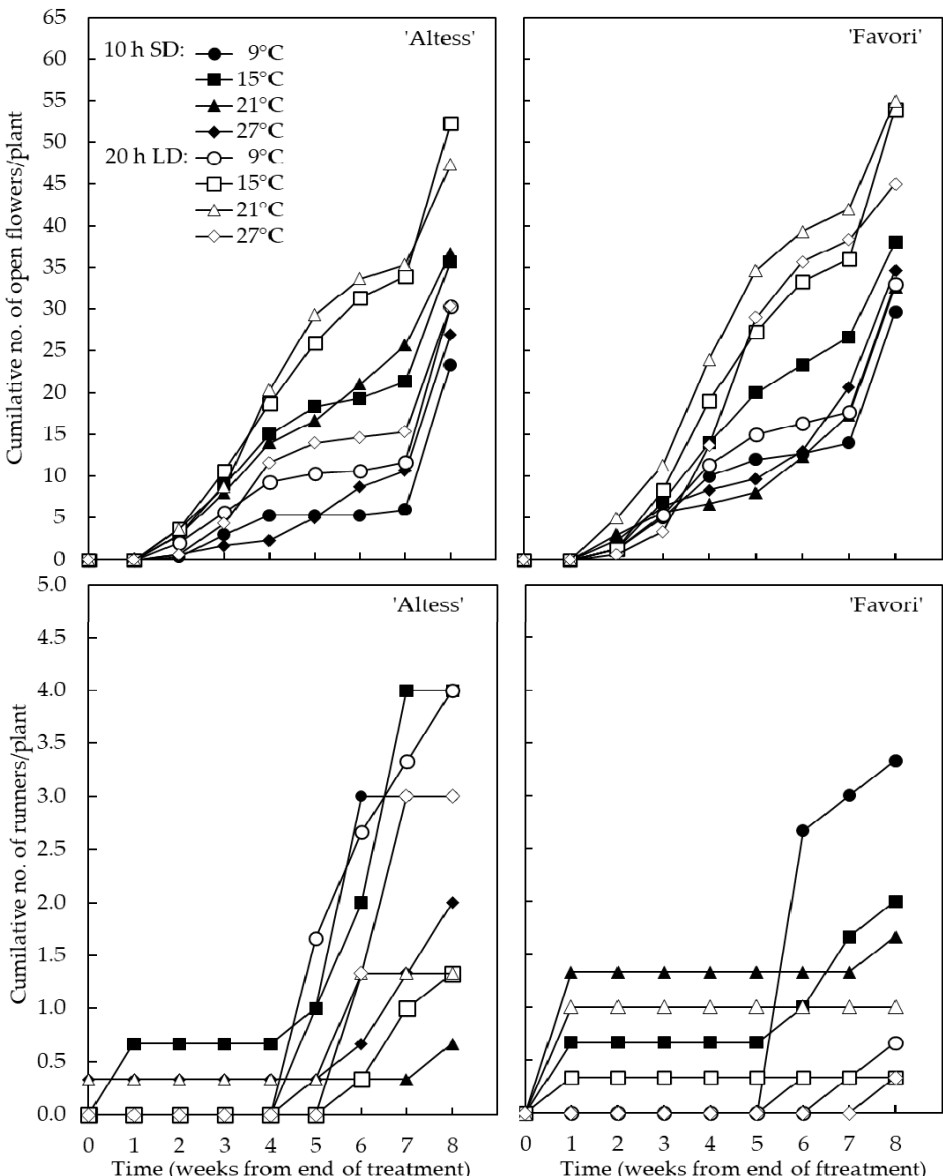

**Figure 4.** Time courses of cumulative flower (top panel) and runner formation in two EB strawberry cultivars as affected by 4 weeks of temperature and photoperiod pre-conditioning as indicated, followed by further raising under natural outdoor conditions from 12 September until 24 October. The plants were forced for 8 weeks in a greenhouse in 20-h LD at 20 °C. The data are means of three replicates, each with one plant of each cultivar.

### 3.4. Yield Performance of the Preconditioned Plants

The highest berry yield of 1.592 g/plant was recorded in 'Favori' plants preconditioned in LD at 15 °C. 'Favori' also had the highest total berry yield across all temperatures and daylengths, whereas 'Murano' had the lowest total yields (Table 1). In general, the yields were highest at intermediate temperatures (15–21 °C), and lowest at 27 °C, while the effect of photoperiod varied with cultivar and temperature. For 'Altess', the highest total yields were obtain in plants preconditioned in SD across the range of temperatures. For 'Favori', the highest total yields were obtained in plants preconditioned in LD at temperatures up to 21 °C, while at 27 °C, the yields were higher in SD. For 'Murano', the highest total yields were obtained in LD at intermediate temperatures, while in SD at temperatures of 9 °C and 27 °C.

**Table 1.** Effects of preconditioning temperature and photoperiod on total berry yield and yield components of three EB strawberry cultivars when cropped on a tabletop system in high plastic tunnels in the year after treatment. Data are means ± SD of three replications, each with four plants of each cultivar. Plants were harvested from 8 July to 2 October 2019.

| Cultivar | Temp. (°C) | Yield (g/Plant) | | Berry Weight (g) | | Berries Plant$^{-1}$ | |
|---|---|---|---|---|---|---|---|
| | | Photoperiod (h) | | | | | |
| | | 10 | 20 | 10 | 20 | 10 | 20 |
| Altess | 9 | 1276.5 ± 147.4 | 1235.6 ± 82.6 | 20.2 ± 0.6 | 18.2 ± 1.0 | 63.3 ± 7.5 | 67.9 ± 5.1 |
| | 15 | 1266.2 ± 211.2 | 1261.3 ± 53.9 | 18.2 ± 1.3 | 16.8 ± 0.6 | 70.2 ± 16.2 | 75.1 ± 5.1 |
| | 21 | 1204.9 ± 98.1 | 1143.7 ± 331.7 | 18.6 ± 0.4 | 15.9 ± 0.7 | 64.8 ± 5.3 | 71.8 ± 20.1 |
| | 27 | 1134.7 ± 204.8 | 1054.7 ± 352.1 | 17.6 ± 0.7 | 15.6 ± 0.4 | 64.4 ± 12.3 | 67.6 ± 22.9 |
| | Mean | 1220.6 | 1173.8 | 18.7 | 16.6 | 65.7 | 70.6 |
| Favori | 9 | 1318.5 ± 12.9 | 1320.1 ± 20.5 | 18.0 ± 1.2 | 18.3 ± 0.3 | 73.3 ± 4.6 | 72.3 ± 0.8 |
| | 15 | 1193.9 ± 99.3 | 1592.4 ± 203.0 | 18.1 ± 0.7 | 17.0 ± 1.1 | 66.0 ± 7.9 | 93.4 ± 6.6 |
| | 21 | 1216.6 ± 124.7 | 1280.6 ± 263.8 | 17.8 ± 0.9 | 15.6 ± 0.3 | 68.6 ± 8.8 | 82.3 ± 17.8 |
| | 27 | 1149.6 ± 234.1 | 1105.1 ± 439.2 | 17.1 ± 0.9 | 13.5 ± 0.3 | 67.8 ± 17.4 | 81.5 ± 32.3 |
| | Mean | 1219.6 | 1324.5 | 17.8 | 16.1 | 68.9 | 82.4 |
| Murano | 9 | 1063.0 ± 177.3 | 864.5 ± 634.2 | 19.0 ± 1.9 | 14.9 ± 3.2 | 56.4 ± 12.7 | 54.0 ± 30.8 |
| | 15 | 1065.0 ± 296.6 | 1219.1 ± 107.4 | 19.0 ± 0.4 | 17.6 ± 0.6 | 56.1 ± 16.9 | 69.2 ± 5.9 |
| | 21 | 848.2 ± 247.7 | 1114.1 ± 192.5 | 17.0 ± 1.3 | 17.4 ± 1.0 | 50.4 ± 16.3 | 64.0 ± 8.2 |
| | 27 | 1065.7 ± 180.5 | 589.5 ± 116.4 | 18.7 ± 0.6 | 14.4 ± 1.2 | 57.1 ± 11.0 | 40.8 ± 6.9 |
| | Mean | 1010.0 | 946.8 | 18.5 | 16.1 | 55.0 | 57.0 |
| Probability level of significance (ANOVA) | | | | | | | |
| Source of variation | | | | | | | |
| Temperature (A) | | <0.001 | | ns | | ns | |
| Photoperiod (B) | | <0.001 | | <0.001 | | <0.001 | |
| A × B | | 0.03 | | <0.001 | | ns | |
| Cultivar (C) | | <0.001 | | <0.001 | | 0.001 | |
| C × A | | ns | | ns | | 0.001 | |
| C × B | | ns | | ns | | ns | |
| A × B × C | | ns | | ns | | 0.04 | |

**Table 1.** *Cont.*

| Cultivar | Temperature (°C) | Berries >25 mm (%) | | Unsaleable Berries (g/plant) | |
|---|---|---|---|---|---|
| | | Photoperiod (h) | | | |
| | | 10 | 20 | 10 | 20 |
| Altess | 9 | 99.6 ± 0.2 | 98.6 ± 1.0 | 19.4 ± 2.9 | 32.4 ± 3.6 |
| | 15 | 99.0 ± 0.4 | 98.6 ± 0.4 | 19.3 ± 3.7 | 20.8 ± 4.8 |
| | 21 | 99.2 ± 0.2 | 98.3 ± 0.6 | 12.3 ± 4.8 | 18.0 ± 4.9 |
| | 27 | 98.1 ± 0.5 | 98.0 ± 0.4 | 11.6 ± 3.7 | 14.1 ± 5.7 |
| | Mean | 99.0 | 98.4 | 15.6 | 21.3 |
| Favori | 9 | 99.0 ± 0.4 | 98.6 ± 0.6 | 17.1 ± 3.5 | 8.8 ± 2.0 |
| | 15 | 99.0 ± 0.9 | 98.1 ± 0.8 | 14.3 ± 3.9 | 9.4 ± 2.9 |
| | 21 | 98.2 ± 0.9 | 97.0 ± 0.4 | 25.4 ± 8.6 | 8.4 ± 2.7 |
| | 27 | 97.6 ± 0.3 | 94.8 ± 0.8 | 16.8 ± 2.1 | 22.4 ± 3.1 |
| | Mean | 98.5 | 97.1 | 18.4 | 12.3 |
| Murano | 9 | 98.2 ± 0.6 | 94.4 ± 5.8 | 28.2 ± 9.2 | 21.0 ± 2.7 |
| | 15 | 98.8 ± 0.3 | 97.2 ± 0.9 | 25.8 ± 11.1 | 19.9 ± 9.9 |
| | 21 | 96.6 ± 1.4 | 96.5 ± 0.7 | 8.9 ± 3.1 | 15.0 ± 2.3 |
| | 27 | 97.1 ± 1.8 | 95.4 ± 0.8 | 9.9 ± 5.7 | 2.3 ± 1.1 |
| | Mean | 97.7 | 95.9 | 18.2 | 12.6 |
| Probability level of significance (ANOVA) | | | | | |
| Source of variation | | | | | |
| Temperature (A) | | ns | | ns | |
| Photoperiod (B) | | 0.005 | | 0.003 | |
| A × B | | 0.01 | | ns | |
| Cultivar (C) | | <0.001 | | 0.007 | |
| C × A | | ns | | ns | |
| C × B | | ns | | ns | |
| A × B × C | | ns | | ns | |

ns, not significant.

The yield components, berry weight (size) and number of berries per plant, varied in an inverse relationship in all cultivars, the former being generally enhanced by SD and the latter by LD preconditioning at all temperatures (some variation in 'Murano', however) (Table 1). Berry weight decreased with increasing temperature in both photoperiods, but due to a highly significant interaction of photoperiod and temperature, the main effect of temperature was not statistically significant. The number of berries per plant varied significantly between the cultivars, being lower in 'Murano' than in the other two cultivars (highly significant interaction between cultivar and temperature).

As shown in Figure 5, the temporal distribution of the berry production and the size of the fruiting flushes differed widely between cultivars and treatments. There was a clear relationship between the size of the first fruit flush and the size and temporal distribution of the rest of the crop. A large and concentrated first fruit flush was always associated with a marked subsequent off period with little or no fruit. In 'Altess' and particularly in 'Favori', the severity and duration of the off period increased markedly with increasing preconditioning temperature. In fact, 'Favori' plants preconditioned in LD at 27 °C, recurrent flowering and fruiting never fully recovered during the season. In 'Murano', which generally had relatively small first fruit flushes and a more even distribution of the crop during the harvest season, this effect was only present in plants preconditioned at 27 °C in SD. Generally, smaller first fruiting flushes led to more stable berry yields during the rest of the season. For all three cultivars, the first ripe fruits appeared after 5 weeks of cultivation in the polytunnel. Plants preconditioned at 9 °C always had a small first flush regardless of photoperiod in all cultivars, but had relatively constant berry production during the cropping season (Figure 5).

The time courses of runner production during the cropping season shown in Figure 6, demonstrating a general declining seasonal trend in all cultivars. A large share of the runners were produced before week 28 when the berry harvest started, and the majority appeared during the first half of the season. 'Favori' produced less runners than the other two cultivars, and in all cultivars the highest number of runners was produced in plants preconditioned in SD (Table 2). At intermediate to high temperatures, runners and flowers were produced in parallel in all cultivars during weeks 28 to 31, while at 9 °C, there was an opposite trend between runner and flower production in 'Murano' and 'Altess'. For 'Favori', there was a decreasing trend for both runner and flower production during the same period.

Some measures of plant architecture at the end of the harvest season are presented in Table 2. All the parameters measured varied significantly between the cultivars. The growth-related parameters (plant height, crowns, leaves and runners per plant), as well as plant fresh weight and inflorescences per plant, were always enhanced by SD and were highest in 'Favori'. The main effect of temperature was significant only for plant height, inflorescences per plant and flowers/fruits not harvested. In all cultivars, the total number of inflorescences were usually highest at intermediate temperatures (21 °C and 15 °C) in both daylengths. The number of flowers and fruits that did not reach maturity before the harvest was terminated on 8 October was significantly higher in plants preconditioned in SD at intermediate temperatures (Table 2).

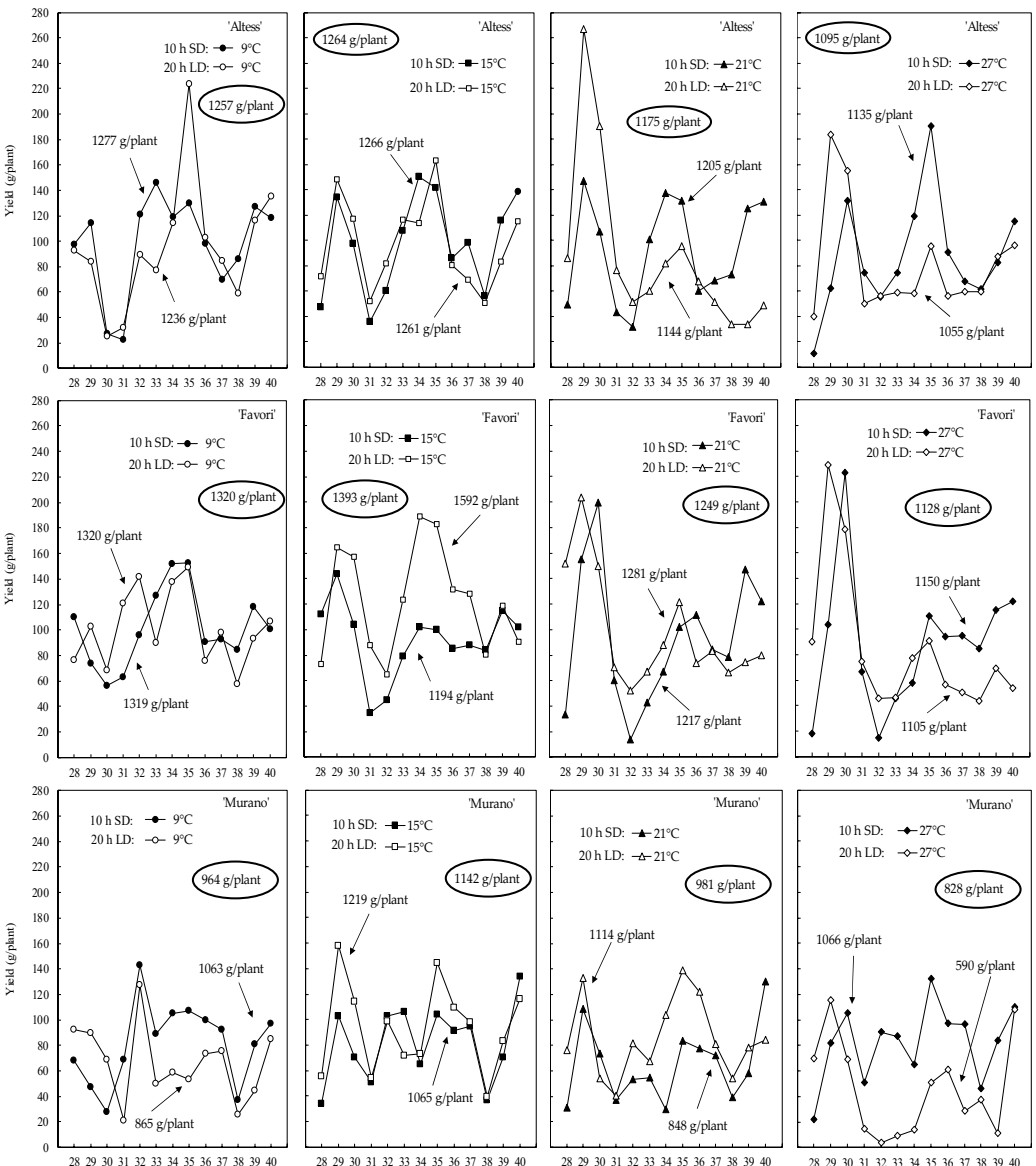

**Figure 5.** Time courses of weekly berry yield of three EB strawberry cultivars as affected by 4 weeks of preconditioning at temperatures of 9 °C, 15 °C, 21 °C, 27 °C and 10 h or 20 h photoperiods, as indicated, followed by further raising under natural outdoor conditions from 12 September until 24 October 2018. The plants were cropped in a plastic tunnel after overwintering in cold storage at −1.5 °C from 24 October 2018 until 5 May 2019. The data are means of three replications with four plants each. Values in the ovals represent the mean yield for both photoperiods at each temperature.

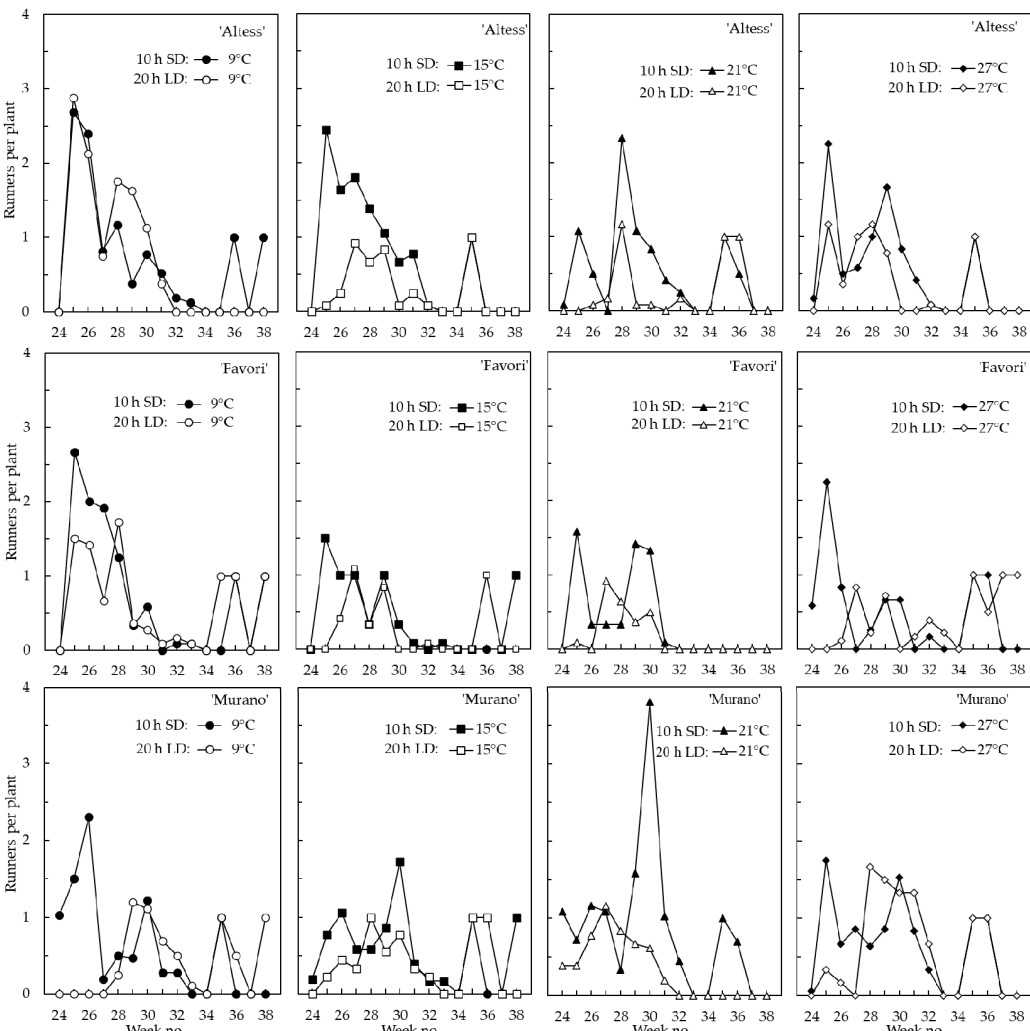

**Figure 6.** Time courses of weekly runner production of three EB strawberry cultivar as affected by 4 weeks of preconditioning at temperatures of 9 °C, 15 °C, 21 °C, 27 °C and 10 h or 20 h photoperiods, followed by further raising under natural outdoor conditions from 12 September until 24 October 2018. The plants were cropped in a plastic tunnel after overwintering in cold storage at −1.5 °C from 24 October 2018 until 5 May 2019. The data are means of three replications with four plants each.

**Table 2.** Effects of raising temperature and photoperiod on plant architecture in three EB strawberry cultivars grown on a tabletop system in a plastic high tunnel in the year after treatment. Data are means ± SD of three replicates, each with four plants of each cultivar.

| Cultivar | Temp. (°C) | Plant Height (cm) | | Crowns Plant$^{-1}$ | | Leaves Plant$^{-1}$ | | Runners Plant$^{-1}$ * | |
|---|---|---|---|---|---|---|---|---|---|
| | | Photoperiod (h) | | | | | | | |
| | | 10 | 20 | 10 | 20 | 10 | 20 | 10 | 20 |
| Altess | 9 | 28.0 ± 2.3 | 28.5 ± 0.7 | 4.0 ± 1.0 | 4.1 ± 0.8 | 31.4 ± 6.3 | 37.0 ± 2.8 | 5.6 ± 1.3 | 7.8 ± 1.7 |
| | 15 | 28.0 ± 1.4 | 25.9 ± 2.8 | 4.5 ± 1.2 | 4.4 ± 0.8 | 32.8 ± 8.8 | 35.8 ± 0.5 | 8.4 ± 1.5 | 2.9 ± 0.4 |
| | 21 | 27.0 ± 1.8 | 23.9 ± 2.1 | 3.9 ± 0.1 | 3.0 ± 0.6 | 35.4 ± 4.2 | 24.4 ± 6.2 | 7.1 ± 1.8 | 2.1 ± 0.7 |
| | 27 | 27.1 ± 4.1 | 23.1 ± 3.7 | 4.6 ± 0.1 | 2.9 ± 1.2 | 32.5 ± 1.4 | 26.7 ± 8.1 | 6.6 ± 1.3 | 4.4 ± 1.8 |
| | Mean | 27.5 | 25.4 | 4.3 | 3.6 | 33.2 | 30.9 | 7.0 | 4.3 |
| Favori | 9 | 29.8 ± 3.4 | 29.8 ± 0.6 | 5.9 ± 0.8 | 3.8 ± 0.4 | 38.9 ± 7.2 | 35.8 ± 1.9 | 6.9 ± 1.8 | 4.6 ± 1.9 |
| | 15 | 33.2 ± 2.7 | 33.2 ± 3.2 | 5.3 ± 1.8 | 6.3 ± 0.7 | 41.6 ± 10.6 | 51.8 ± 6.6 | 4.3 ± 2.2 | 2.3 ± 1.7 |
| | 21 | 32.8 ± 3.3 | 26.0 ± 1.1 | 5.5 ± 0.3 | 5.1 ± 0.5 | 57.8 ± 2.6 | 41.7 ± 6.4 | 5.1 ± 0.4 | 2.5 ± 1.1 |
| | 27 | 31.3 ± 3.6 | 22.9 ± 2.9 | 7.3 ± 0.8 | 4.3 ± 1.7 | 55.3 ± 7.7 | 31.2 ± 9.3 | 4.6 ± 1.5 | 2.6 ± 1.3 |
| | Mean | 31.8 | 28.0 | 6.0 | 4.8 | 48.4 | 40.1 | 5.2 | 3.0 |
| Murano | 9 | 23.1 ± 2.3 | 19.6 ± 3.8 | 4.8 ± 1.4 | 4.2 ± 2.0 | 30.6 ± 4.5 | 35.4 ± 17.4 | 5.5 ± 0.9 | 5.1 ± 1.1 |
| | 15 | 24.4 ± 2.5 | 23.6 ± 3.2 | 4.8 ± 1.6 | 4.0 ± 1.3 | 37.3 ± 8.6 | 28.3 ± 8.5 | 5.4 ± 0.5 | 3.4 ± 1.1 |
| | 21 | 26.9 ± 2.7 | 21.8 ± 1.9 | 6.3 ± 1.4 | 3.6 ± 0.5 | 44.1 ± 9.4 | 26.1 ± 7.1 | 10.1 ± 3.5 | 4.3 ± 1.5 |
| | 27 | 23.6 ± 2.5 | 17.2 ± 5.8 | 7.3 ± 1.4 | 5.2 ± 2.0 | 43.9 ± 19.3 | 33.3 ± 17.4 | 6.9 ± 2.3 | 6.8 ± 3.6 |
| | Mean | 24.5 | 20.5 | 5.8 | 4.2 | 39.0 | 30.8 | 7.0 | 4.9 |
| Probability level of significance (ANOVA) | | | | | | | | | |
| Source of variation | | | | | | | | | |
| Temperature (A) | | <0.010 | | ns | | ns | | ns | |
| Photoperiod (B) | | <0.001 | | 0.042 | | 0.038 | | <0.001 | |
| A × B | | 0.001 | | ns | | 0.006 | | 0.019 | |
| Cultivar(C) | | 0.022 | | 0.044 | | 0.010 | | 0.008 | |
| C × A | | ns | | ns | | ns | | ns | |
| C × B | | ns | | ns | | ns | | ns | |
| A × B × C | | ns | | ns | | ns | | ns | |

**Table 2.** *Cont.*

| Cultivar | Temp. (°C) | Plant FW (g) | | Infloresc. Plant$^{-1}$ * | | Flowers/Fruits not Harvested | |
|---|---|---|---|---|---|---|---|
| | | Photoperiod (h) | | | | | |
| | | 10 | 20 | 10 | 20 | 10 | 20 |
| Altess | 9 | 287.1 ± 21.1 | 295.2 ± 54.0 | 12.2 ± 0.1 | 12.2 ± 2.6 | 70.7 ± 11.3 | 78.3 ± 14.5 |
| | 15 | 267.6 ± 30.0 | 211.3 ± 5.5 | 12.8 ± 1.6 | 14.5 ± 0.7 | 85.1 ± 14.3 | 62.9 ± 6.0 |
| | 21 | 277.0 ± 23.9 | 146.6 ± 43.1 | 14.3 ± 0.6 | 11.3 ± 2.3 | 81.8 ± 14.6 | 61.3 ± 13.9 |
| | 27 | 261.2 ± 15.6 | 158.1 ± 56.7 | 13.4 ± 0.8 | 11.3 ± 1.7 | 74.3 ± 12.3 | 61.5 ± 17.0 |
| | Mean | 271.9 | 202.8 | 13.3 | 12.3 | 78.6 | 66.0 |
| Favori | 9 | 297.3 ± 61.4 | 262.3 ± 16.3 | 13.6 ± 3.1 | 11.3 ± 0.7 | 63.5 ± 20.2 | 52.8 ± 13.7 |
| | 15 | 320.6 ± 66.3 | 341.9 ± 62.2 | 16.3 ± 3.4 | 17.4 ± 2.4 | 88.6 ± 43.8 | 71.8 ± 26.5 |
| | 21 | 411.9 ± 34.8 | 231.6 ± 35.8 | 19.4 ± 1.3 | 18.6 ± 4.5 | 117.7 ± 6.5 | 76.6 ± 10.4 |
| | 27 | 385.2 ± 56.0 | 179.7 ± 42.2 | 18.3 ± 3.4 | 13.7 ± 1.5 | 112.3 ± 31.7 | 66.1 ± 5.5 |
| | Mean | 353.8 | 253.9 | 16.9 | 15.2 | 95.5 | 66.8 |
| Murano | 9 | 185.0 ± 14.8 | 157.3 ± 94.2 | 12.6 ± 1.9 | 9.5 ± 3.5 | 76.3 ± 17.0 | 66.0 ± 24.1 |
| | 15 | 228.6 ± 41.5 | 157.8 ± 57.5 | 14.6 ± 3.9 | 11.8 ± 1.7 | 114.2 ± 15.9 | 67.7 ± 14.9 |
| | 21 | 242.8 ± 46.6 | 161.9 ± 35.4 | 17.6 ± 1.7 | 12.7 ± 2.7 | 128.2 ± 18.1 | 59.3 ± 10.4 |
| | 27 | 211.3 ± 30.1 | 163.5 ± 68.3 | 14.6 ± 6.2 | 15.2 ± 6.3 | 98.7 ± 41.0 | 112.5 ± 53.5 |
| | Mean | 216.9 | 160.1 | 14.8 | 12.3 | 104.3 | 76.4 |
| Probability level of significance (ANOVA) | | | | | | | |
| Source of variation | | | | | | | |
| Temperature (A) | | ns | | 0.019 | | 0.042 | |
| Photoperiod (B) | | <0.001 | | ns | | 0.022 | |
| A × B | | 0.002 | | ns | | ns | |
| Cultivar(C) | | <0.001 | | 0.044 | | ns | |
| C × A | | ns | | ns | | ns | |
| C × B | | ns | | ns | | ns | |
| A × B × C | | ns | | ns | | ns | |

ns, not significant. * Total number of runners and inflorescences produced during the cropping season.

## 4. Discussion

### 4.1. Flowering and Runnering in the Phytotron

The number of runners and flowers produced during the 4-week preconditioning period (Figure 2) was merely attributed to the previous environmental history of the stock plants from which the runners were taken. Accordingly, their time of emergence was enhanced by increasing temperatures during the treatment period. It is well documented that, due to their perpetual flowering nature, the runnering capacity of EB strawberry cultivars is generally low [2,7]. Although Rivero et al. [2] and Sønsteby et al. [4] showed that 'Favori' produced relatively few runners, the cultivar had more runners than 'Altess' and 'Murano' in the present experiment. Obviously, these deviations were due to the different environmental prehistory of the stock plants. As generally found in EB strawberry cultivars [2,5], flowering and runnering varied in an inverse manner and a rapid increase in flower emergence took place in all cultivars as runner formation levelled off (Figure 2).

### 4.2. Flowering Potential of the Preconditioned Plants

An inverse relationship between flowering and runnering was demonstrated in all cultivars and most clearly in 'Favori', where a rapid increase in the number of flowers after week 3 of forcing was associated with cessation of runner production in all treatments (Figure 3). A similar situation was revealed in the LD-treated plants of the other cultivars, although a few runners emerged in the sparsely flowering SD-treated plants. In all cultivars, flowering was enhanced by LD and increasing temperatures (except for 'Murano' at 15 °C), whereas SD strongly delayed flowering at high temperatures, especially in 'Favori' and 'Murano'. On the other hand, runner formation was enhanced by SD over the same intermediate to high temperature range. All these results concur with previous results for EB cultivars in general [8,9] as well as for 'Favori' and other modern EB cultivars [2,4,10,11]. Furthermore, a new flush of flowers emerged after a time span of approximately seven weeks, which had apparently been initiated after the plants had been transferred to the LD and relatively high temperature forcing conditions. This second flush of flowering was less pronounced in 'Murano' than in 'Favori' and 'Altess' (Figure 3).

### 4.3. Plant Flowering Potential in Late Autumn

The flowering and runnering performance of the plants after completion of the raising season was in many ways comparable with those of plants forced immediately after completion of the 4-week preconditioning treatment (cf. Figures 3 and 4). The flower-promoting effects of LD and high temperature preconditioning were still pronounced, and the inverse relationship between flowering and runnering also remained much the same. However, both flowers and runners started to emerge about one week earlier in the later forced plants, and in all treatments, the emergence of flowers levelled off after approximately six weeks of forcing. This was associated with the emergence of a few runners in plants with sparse flowering. Furthermore, as in the directly forced plants, a new flush of flowers emerged after seven weeks of forcing. Since the increase in number of flowers was identical in all preconditioned plants groups, these flowers were clearly initiated during the forcing treatment, as previously suggested for the directly forced plants.

These results demonstrated that recurrent floral initiation readily takes place when previously induced plants with advanced flower primordia are transferred back to inductive conditions. The critical point seemed to be whether the plants had become dormant or not and whether they were exposed to dormancy-breaking chilling. In the present experiment, the plants were exposed to daily mean temperatures ranging from 5–15 °C for six weeks, and the levelling in the number of open flowers (in Figure 4) indicated that plants from all treatments had been at the verge of becoming dormant when the forcing started. This concurred with the results of Rivero et al. (2021) [2] who found that low temperature (6 °C) did not induce dormancy in 'Favori' plants regardless of daylength conditions, while they became dormant after 10 weeks (but not 5 weeks) of SD exposure at 16 and 26 °C. It is also clear that the present temperature conditions did not bring about the delay of

recurrent floral initiation that results from long-term chilling at sub-zero temperatures (cf. Gallace et al. [5]).

Another interesting finding was that the number of flowers emerging did not continue to increase when the pretreated plants were grown further under natural outdoor conditions for another six weeks before forcing. In fact, the number of emerging flowers and runners decreased somewhat, even when compensating for the 2-week shorter forcing period (cf. Figures 3 and 4). This indicated that some flower primordia aborted during autumn and suggested a limit to how many viable flowers the plants could accumulate and support until flowering and fruiting. This was contrary to the stabilization of flowering by low temperature that we expected.

### 4.4. Yield Performance of the Preconditioned Plants

A rather short exposure to varying temperature and photoperiod conditions during the raising of the plants in one year had a remarkable effect on the flowering and fruiting pattern in the following year. The highest yields were generally obtained in plants exposed to LD at intermediate temperatures of 15–21 °C. This temperature range has previously been reported as highly effective for LD-induced flower bud formation in EB cultivars [2,8–10] and was also found to be optimal for photosynthesis in these cultivars [12]. As previously reported by Melis [6], treatments that produced a large first fruiting flush were generally associated with a long off period with little flowering and fruiting. This was particularly pronounced in plants preconditioned in LD at 21 and 27 °C, conditions that are optimal for flower initiation in EB strawberries, whereas treatments that yielded smaller first fruit flushes gave a more balanced and evenly distributed harvest during the season (Figure 5). It is interesting to note that the time lapse between fruiting flushes was usually six to seven weeks, the same time as found for recurrent flowering in greenhouse-forced plants.

As flowers and developing fruits are strong sinks for photosynthates, Sønsteby et al. [4,7] found that a heavy fruit load not only constrained the growth of developing fruits, but also repressed and delayed recurrent initiation of successive flushes, as demonstrated by Melis [6]. While this trend was pronounced in 'Favori' and 'Altess' it was less marked in 'Murano', as also found by Sønsteby et al. [4]. Thus, the present results concurred with previous reports, confirming that the high-yielding 'Favori' was more vulnerable to this phenomenon than the lower yielding 'Murano'. Because of the long off periods associated with large first flushes in plants preconditioned at high temperature, the total fruit yield over the entire harvest season was generally highest in plants preconditioned at 15 °C (at 21 °C in 'Murano'), while it always declined at higher temperatures. As discussed by Sønsteby et al. [7], continuous and excessive flower initiation in LD at high temperatures also resulted in reduced leaf canopy and plant weight. Similarly, Rivero et al. [2] found that 'Favori' was susceptible to overproduction of flowers at optimal flowering conditions, and that this could have detrimental consequences for maintenance of the leaf canopy and perpetual flower initiation. Therefore, as discussed by Sønsteby et al. [4], the magnitude of the first crop flush may have to be compromised in order to optimize total yield and the seasonal distribution of the crop.

The number of runners produced during the cropping season was relatively low, although it increased significantly after winter chilling, as reported by Gallace et al. [5]. The number was lowest in 'Favori', and in all cultivars, a large share was produced during the first five weeks before the fruit harvest started (Figure 6).

## 5. Conclusions

The results of the present study demonstrated that a brief exposure of 4 weeks to varying temperature and photoperiod conditions during plant raising had a strong effect on the instant flowering potential of the plants and a remarkable long-term effect on the yield and temporal distribution of the harvest in the following year in the EB strawberry cultivars Altess, Favori and Murano. The responses were determined by a pronounced interaction of temperature and photoperiod. As previously reported for these and other

EB cultivars, the perpetual flowering proved to be source-limited, since large first fruiting flushes repressed and delayed successive flushes and strongly reduced plant leaf canopy. Accordingly, the size of the first fruiting flush must be compromised in order to optimize total yield and the seasonal distribution of the harvest. Through proper manipulation of the light and temperature environment during plant raising, it seems feasible to design tailor-made plant production programs for high and continuous yields.

**Author Contributions:** Conceptualization and methodology, R.R., S.F.R., O.M.H. and A.S.; data curation software, A.S., O.M.H. and R.R.; validation, R.R., S.F.R., O.M.H. and A.S.; formal analysis, A.S. and R.R.; investigation, R.R.; resources, A.S. and S.F.R.; writing—original draft preparation, R.R.; writing—review and editing, R.R., S.F.R., O.M.H. and A.S.; visualization, R.R., S.F.R., O.M.H. and A.S.; supervision, R.R., S.F.R., O.M.H. and A.S.; project administration, A.S. and S.F.R.; funding acquisition, A.S. and S.F.R. All authors have read and agreed to the published version of the manuscript.

**Funding:** This research was funded by the PhD programme at the Norwegian University of Life Sciences, and the Norwegian Agricultural Agreement Research Fund/Foundation for Research Levy on Agricultural Products, grant number 280608 and Grofondet, grant number 170008.

**Institutional Review Board Statement:** Not Applicable.

**Informed Consent Statement:** Not Applicable.

**Data Availability Statement:** The plant and yield data are available upon request from the corresponding author.

**Acknowledgments:** We thank Unni M. Roos, Signe Hansen and Kari Grønnerød for excellent technical assistance.

**Conflicts of Interest:** The authors declare no conflict of interest.

## Appendix A

**Table 1.** Appearances of flowers and runners in three EB strawberry cultivars during the 4-week preconditioning treatment in the phytotron. The data are means ± SD of three replicates, each with 10 plants of each cultivar.

| Cultivar | Temperature (°C) | Flowers Plant$^{-1}$ | | Runners Plant$^{-1}$ | |
|---|---|---|---|---|---|
| | | Photoperiod (h) | | | |
| | | 10 | 20 | 10 | 20 |
| Altess | 9 | 1.0 ± 0.4 | 1.3 ± 0.4 | 0.1 ± 0.0 | 0.1 ± 0.0 |
| | 15 | 2.5 ± 0.8 | 4.6 ± 0.4 | 0.5 ± 0.2 | 0.5 ± 0.1 |
| | 21 | 5.5 ± 2.4 | 8.0 ± 1.0 | 0.1 ± 0.0 | 0.5 ± 0.2 |
| | 27 | 8.6 ± 1.3 | 8.9 ± 2.8 | 0.3 ± 0.1 | 0.5 ± 0.1 |
| | Mean | 4.4 | 5.7 | 0.2 | 0.4 |
| Favori | 9 | 0.7 ± 0.1 | 0.7 ± 0.1 | 0.0 ± | 0.0 ± 0.0 |
| | 15 | 3.6 ± 0.0 | 5.9 ± 1.2 | 0.4 ± | 0.6 ± 0.1 |
| | 21 | 6.8 ± 0.9 | 10.6 ± 1.4 | 0.6 ± | 0.9 ± 0.3 |
| | 27 | 4.0 ± 0.9 | 7.6 ± 1.8 | 1.5 ± | 1.2 ± 0.2 |
| | Mean | 3.8 | 6.2 | 0.6 | 0.7 |
| Murano | 9 | 3.8 ± 0.9 | 7.1 ± 1.5 | 0.0 ± 0.0 | 0.0 ± 0.0 |
| | 15 | 5.9 ± 1.7 | 10.0 ± 2.5 | 0.1 ± 0.0 | 0.2 ± 0.0 |
| | 21 | 10.0 ± 3.2 | 11.7 ± 1.3 | 0.4 ± 0.0 | 0.4 ± 0.1 |
| | 27 | 13.6 ± 5.4 | 16.9 ± 1.4 | 0.2 ± 0.0 | 0.1 ± 0.0 |
| | Mean | 8.3 | 11.4 | 0.2 | 0.2 |
| Probability level of significance (ANOVA) | | | | | |
| Source of variation | | | | | |
| Temperature (A) | | <0.001 | | <0.001 | |
| Photoperiod (B) | | 0.003 | | ns | |
| A × B | | ns | | ns | |
| Cultivar (C) | | <0.001 | | <0.001 | |
| C × A | | 0.002 | | <0.001 | |
| C × B | | ns | | ns | |
| A × B × C | | ns | | ns | |

ns, not significant.

**Table 2.** Effects of temperature and photoperiod during plant raising on growth and flowering of three EB strawberry cultivars as assessed by 10 weeks of forcing started immediately after 4 weeks of preconditioning, and by 8 weeks of forcing started in late autumn after completion of plant raising (preconditioning + outdoor treatment). In both cases the plants were forced in a heated greenhouse maintained at 20 h light and 20 °C. The data are means ±SD of three replicates, each with 3 or 1 plant respectively, of each cultivar.

| Cultivar | Temperature (°C) | After Preconditioning | | | | In Late Autumn | | | |
| | | Flowers Plant$^{-1}$ | | Runners Plant$^{-1}$ | | Flowers Plant$^{-1}$ | | Runners Plant$^{-1}$ | |
| | | Photoperiod (h) | | | | | | | |
| | | 10 | 20 | 10 | 20 | 10 | 20 | 10 | 20 |
|---|---|---|---|---|---|---|---|---|---|
| Altess | 9 | 42.9 ± 6.9 | 56.9 ± 13.0 | 0.3 ± 0.0 | 0.7 ± 0.3 | 23.3 ± 6.0 | 30.3 ± 4.0 | 3.0 ± 1.0 | 4.0 ± 1.0 |
| | 15 | 63.0 ± 8.3 | 87.8 ± 23.0 | 1.3 ± 0.2 | 0.5 ± 0.0 | 35.7 ± 9.1 | 52.3 ± 14.0 | 4.0 ± 0.0 | 2.0 ± 0.0 |
| | 21 | 58.8 ± 1.9 | 83.3 ± 23.0 | 1.1 ± 0.6 | 0.7 ± 0.0 | 36.7 ± 7.2 | 47.3 ± 8.5 | 1.0 ± 0.0 | 2.0 ± 0.4 |
| | 27 | 75.2 ± 14.5 | 67.3 ± 21.2 | 1.3 ± 0.6 | 0.4 ± 0.1 | 27.0 ± 3.5 | 30.3 ± 7.2 | 2.0 ± 0.8 | 3.0 ± 1.7 |
| | Mean | 60.0 | 73.8 | 1.1 | 0.6 | 30.7 | 40.1 | 2.6 | 2.9 |
| Favori | 9 | 49.9 ± 8.2 | 54.9 ± 6.9 | 0.5 ± 0.1 | 0.3 ± 0.0 | 29.7 ± 2.3 | 33.0 ± 7.2 | 3.3 ± 1.1 | 2.0 ± 0.0 |
| | 15 | 91.1 ± 3.7 | 99.9 ± 14.6 | 0.3 ± 0.0 | 0.3 ± 0.0 | 38.0 ± 7.0 | 54.0 ± 4.4 | 2.0 ± 0.1 | 1.0 ± 0.0 |
| | 21 | 81.7 ± 5.5 | 109.2 ± 27.1 | 0.0 ± 0.0 | 0.5 ± 0.0 | 32.7 ± 4.9 | 55.0 ± 7.9 | 1.7 ± 0.2 | 1.5 ± 0.7 |
| | 27 | 79.8 ± 16.6 | 92.8 ± 5.4 | 0.4 ± 0.1 | 0.7 ± 0.0 | 34.7 ± 7.2 | 45.0 ± 7.0 | 1.0 ± 0.0 | 1.0 ± 0.0 |
| | Mean | 75.6 | 89.2 | 0.4 | 0.4 | 33.8 | 46.8 | 2.2 | 1.4 |
| Murano | 9 | 43.5 ± 13.7 | 19.0 ± 8.5 | 2.3 ± 1.0 | 0.0 ± 0.0 | - | - | - | - |
| | 15 | 47.6 ± 1.9 | 59.7 ± 22.7 | 2.7 ± 0.5 | 0.3 ± 0.0 | - | - | - | - |
| | 21 | 47.0 ± 13.1 | 52.7 ± 18.3 | 4.7 ± 2.4 | 3.0 ± 0.0 | - | - | - | - |
| | 27 | 33.8 ± 6.8 | 44.8 ± 21.9 | 3.8 ± 1.1 | 1.3 ± 0.5 | - | - | - | - |
| | Mean | 43.8 | 46.3 | 3.5 | 1.5 | - | - | - | - |
| Probability level of significance (ANOVA) | | | | | | | | | |
| Source of variation | | | | | | | | | |
| Temperature (A) | | 0.008 | | ns | | 0.004 | | 0.041 | |
| Photoperiod (B) | | 0.03 | | <0.001 | | 0.001 | | ns | |
| A × B | | ns | | ns | | ns | | 0.030 | |
| Cultivar (C) | | <0.001 | | <0.001 | | ns | | ns | |
| C × A | | ns | | 0.04 | | ns | | ns | |
| C × B | | ns | | <0.001 | | ns | | ns | |
| A × B × C | | ns | | ns | | ns | | ns | |

ns, not significant.

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
