# Peer review of "Effect of Temperature and Photoperiod Preconditioning on Flowering and Yield Performance of Three Everbearing Strawberry Cultivars"

_horticulturae, doi:10.3390/horticulturae8060504_

Round 1
Reviewer 1 Report
Dear authors
The presented graphics are not clear. They should be improved.
Are you presenting your results as graphs and tables?
You have some editing mistakes in lines 399, 401 and 403.
What do phytotron and reviled mean?
Author Response
Dear Reviewer 1
We are grateful for the helpful and useful comments and suggestions by you. Below, please find our point-by-point response to your specific comments.
Comment #1: “The presented graphics are not clear. They should be improved”: Does this refer to the resolution of the graphics? Could you be more specific on what should be improved? If it is the resolution, the printed version of the MS if accepted will be of better quality.
Comment #2: “Are you presenting your results as graphs and tables?”. No. That is only the case for the supplementary tables S1 and S2. The results in Tables 1 and 2, are not presented in any figures.
Comment #3: “You have some editing mistakes in lines 399, 401 and 403”.
Line 399 (now line 417): ‘Fvori’, is amended to ‘Favori’
Line 401 (now line 419): “wth” is amended to “with”
Line 403 (now line 421): “allways” is amended to “always”
Comment #4: A phytotron is a technical greenhouse structure used for studies of interactions between plants and the environment. Such a facility is very useful for studies of plant responses to climatic factors, such as temperature, light, humidity, fertigation, which can be fully controlled and thus provide answers to the effect of each factor and their interactions.
Line 194 (now line 195): “reviled” was amended to “revealed”
Reviewer 2 Report
The MS mainly demonstrated that temperature and photoperiod conditions during plant raising had a strong effect on the instant flowering potential of the plants and a remarkable long-term effect on the yield and temporal distribution of the harvest in the following year in three EB strawberry cultivars, which can provide certain tech reference for strawberry cultivation. There are some details to be considered:
1. Materials and Methods part should be more concise and clearer, maybe use table show the design will be better, current situation is ambiguous.
2. The experiment was conducted as a randomized block design with three replicates with 10 plants of each cultivar in each treatment, so all data in table should analyzed differences with marked letters, i.e. means±SD.
3. Discussion part should explain WHY temperature and photoperiod preconditioning affected flowering, runnering, yield etc., not repeat The Results simply.
Author Response
Dear Reviewer 2,
We are grateful for the helpful and useful comments and suggestions by you. Below, please find our point-by-point response to your specific comments.
Comment #1: “Materials and Methods part should be more concise and clearer, maybe use table show the design will be better, current situation is ambiguous”. We agree that the description is a bit complicated, but we think it is clear and consistent (not ambiguous). We do not quite see how it could be presented as a table, so we prefer to keep it as it stands.
Comment #2: “The experiment was conducted as a randomized block design with three replicates with 10 plants of each cultivar in each treatment, so all data in table should analysed differences with marked letters, i.e. means ±SD”. Means ±SD have been included in all tables.
Comment #3: “Discussion part should explain WHY temperature and photoperiod preconditioning affected flowering, yield and runnering, not repeat the results simply”.
In our opinion, the discussion section covering lines 412-428 of the MS gives a comprehensive explanation on the effect of temperature and photoperiod preconditioning on the flowering, explaining how a heavy fruit load can constrain the continued flowering and yield of the crop. Also confer the statements in lines 439-441 in the conclusion. Otherwise, some results are mentioned in the discussion as a brief introduction of each paragraph for comparison with the results of similar studies in the literature.
Reviewer 3 Report
The author revealed the Flowering time and Yield Performance with the effect of temperature and photoperiod preconditioning on in everbearing strawberry. In general, this manuscript is of special interest, which may broad our understanding of the temperature- and photoperiod-induced flowering and yield performance in strawberry. After I reviewed this ms, my concerns are listed as below:
Major
Is there any change of expression of flowering-related gene, such as FT, FLC and SVP? Or any profiles of DEGs by RNA-seq? Currently, this manuscript is more like a phenotypic data description with very limited information for further following studies.
Author Response
Dear Reviewer 3,
We are grateful for the helpful and useful comments and suggestions by you. Below, please find our point-by-point response to your specific comments.
Comment #1: “Is there any change of expression of flowering-related gene, such as FT, FLC and SVP? Or any profiles of DEGs by RNA-seq?. Currently, this manuscript is more like a phenotypic data description with very limited information for further following studies”. Gene expression analyses were beyond the scope of this study (see the aims of the experiment as stated in the introduction on lines 77-80). Research on the causes of irregular and low yields in EB strawberry production in Europe are highly needed and the present experiment was designed and conducted to shed light on these issues.
Round 2
Reviewer 3 Report
This ms is acceptable. I agree with the author response.
Author Response
Dear reviewer,
We are grateful for the positive response on our amended MS and the helpful comments and suggestions by you.
Kind regards